# Seroprevalence of hepatitis B virus among pregnant women attending Antenatal care in Dilla University Referral Hospital Gedio Zone, Ethiopia; health facility based cross-sectional study

Adugnaw Atnafu Atalay[1], Reta Kassa Abebe[2], Aberash Eifa Dadhi[3], Worku Ketema Bededa[4]*

1 Department of Internal Medicine, College of Health Sciences, St Paul Millennium College, Addis Ababa, Ethiopia, 2 School of Public Health, College of Health Sciences, Dilla University, Dilla, Ethiopia, 3 Department of Midwifery, College of Health Sciences, Hawassa University, Hawassa, Ethiopia, 4 Department of Pediatrics and Child Health, College of Health Sciences, Hawassa University, Hawassa, Ethiopia

* workuketema@gmail.com

**Data Availability Statement:** All relevant data are within the paper and its Supporting information files.

## Abstract

### Introduction

A pregnancy that has been complicated with Hepatitis B virus (HBV) infection results in typical management problems for both the mother and the newborn. One of the universal efforts in tackling the impact of chronic HBV is the prevention of mother-to-child transmission during Antenatal care via prompt screening as the majority of chronic infections globally harbored during this period. Rewarding result have been achieved in reducing this problem at this period of life through maternal screening programs and universal vaccination of infants. This study was aimed at assessing the seroprevalence and associated risk factor of HBV among pregnant women attending Antenatal Care (ANC) in Dilla University Referral Hospital (DURH), Southern Ethiopia.

### Method

A facility- based cross- sectional study was conducted from December 01 to May 30, 2017 among pregnant women attending ANC. A total of 236 pregnant women were included in this study. All Pregnant women who were attending antenatal clinic and were volunteer during the study period were included, whereas those women who were unable to communicate due to any problem, and not volunteer to give informed consent were excluded. Volunteer participants were asked to complete a questionnaire and had offered to test for HBsAg infection. The data was analyzed using SPSS version 20 software. Logistic regression was used to determine the association between dependent and independent variables.

**Funding:** This study was funded by the Ministry of Science and Higher Education and supervision of Dilla University College of Medicine and Health Sciences. The funding organization has no role in the design of study and data collection, analysis, interpretation of the results and preparation of manuscript.

**Competing interests:** The authors have declared that no competing interests exist.

## Results

From 215 pregnant women attending ANC, the prevalence of HBsAg by the rapid test was found to be 11 (5.1%). Among the study participants, 91.1% (215) were tested for HIV antibody during the ANC visit, with the positivity rate of 4.5%. The result showed 1.86% of the study participants who were tested for HIV were also positive for HBsAg. Among those factors affecting the transmission of HBV infection, multiple partners and HIV confection have significant association at P-value less than 0.05.

## Conclusion

The Seropositivity of Hepatitis B Virus among Pregnant Women was found to be significant and hence, routine screening of pregnant mother at Antenatal care for this virus, and subsequent management according to the guideline for both the mother and child is recommended.

## 1. Introduction

Of all viral hepatitis, Hepatitis B Virus (HBV) is the most lethal virus. This virus is oncogenic and is one of the responsible viruses for Hepatocellular Carcinoma. Cirrhosis is another chronic complications of HBV [1–4]

It is 100 times more infectious than Human-Immuno-Virus with an incubation period of 6 weeks to 6 months [2, 5].

The vertical transmission of this virus from mother to child is among the common routes of transmission. The transmission is particularly significant in pregnant women with positive HBeAg, which is a surrogate marker of active replication, and hence infectivity. All the three possible contacts, namely during pregnancy, at the time of delivery, and postnatally while breastfeeding, are the potential times of transmission from infected mother to child [5–8]

HBV infection has both maternal and child complications. These include coagulation abnormalities, accentuation of Postpartum Hemorrhage, Renal failure, stillbirths, Neonatal death, Cirrhosis, and Liver Cancer. One of the successes in combating this virus is the invention of an effective vaccine against it. Hence, universal screening in antenatal visits and vaccinations is a priority for the early intervention and prevention of this deadly virus [6, 9–11].

Screening antenatal women for HBsAg can give an upright prevalence of the disease in a population. [7, 9, 12]. Global data showed that about 780 000 people die every year due to the harmful effects of hepatitis B virus [2, 13, 14]. About 8 to 8.2% of pregnant women in Ethiopia are supposed to be infected by this virus and the country is among the highly affected area by this virus [15, 16]

The recent studies that have done globally and locally still showed the moderate prevalence of HBV among pregnant mother as it has been evidenced by the studies done in Gambia, Ghana, Southern Ethiopia, and Northern Ethiopia with the Seropositivity rates of 9.2%, 7.7%,7.3%, and 9.2% respectively [17–20].

The transmission during infancy and early Prevention of vertical transmission is an important area of universal efforts to reduce the burden of HBV as mother-to-child transmission is responsible for approximately one-half of chronic infections globally. "Although there are guidelines for universal infant HBV vaccination, rates of maternal HBV infection have increased annually by 5.5% since 1998" [10]. The risk of transmission is about 90 percent in

those exposed at birth without vaccination, while the risk is about 20 to 30 percent in those exposed during childhood. Maternal screening programs and universal vaccination of infants have significantly reduced transmission rates [9, 11, 21, 22]. As far as our knowledge goes, there is an inadequacy of data regarding the prevalence of HBsAg among pregnant women in the study area and the study was aimed at assessing the burden of HBsAg among pregnant women with the view of alleviating the disease burden by providing universal screening and vaccination.

## 2. Material and method

### 2.1. Study area and period

After Institutional Review Board (IRB) of Dilla University had approved the ethical clearance, the study was conducted in DURH, Southern Ethiopia, altitude of the city ranges between 2000 and 2500 meters above sea level. DURH is found in Dilla Town, Gedeo zone SNNPR. It is located 360kms from Addis Ababa which is the capital city of Ethiopia and 90km from Hawassa which is capital city of SNNPR. Administratively Dill town is divided in to 3 Sub-cities (SC) and 8 Kebeles (K) and an estimated population size of 82,944. Out of these around 19,136 women are child bearing age group (15–49 years) and there are around 7,029 husband's whose wives are pregnant and 6,487 husband's having children below one years old. In the town there are 1 hospital, 2 health center, 33 private clinic, 6 health posts, 15 pharmacies and 28 Drug stores.

### 2.2. Study design and population

A facility based cross sectional study was conducted among pregnant women attending ANC in DURH, Southern Ethiopia. The source populations were all pregnant women attending ANC in DURH. All Pregnant women who were attending antenatal care clinics in DURH, Southern Ethiopia during the study period.

### 2.3. Eligibility

**2.3.1. Inclusion criteria.** All Pregnant women were attending antenatal clinic in the DURH during the study period and who were volunteering and give informed consent.

**2.3.2. Exclusion criteria.** Those women who were unable to communicate due to any problem.

### 2.4. Sample size

Sample was calculated by taking overall Hepatitis B infection prevalence among a cross-sectional study conducted among pregnant women in Gondar in 2008. Of 209 mothers included in the study, 11 (5.3%) 3.0% level of significance / margin of error [23]. This sample size will be estimated using the formula for calculating sample size for cross sectional study of estimation a single population proportion as described below.

$$n = (Z\alpha/2)^2 P(1 - P)/(d)^2 = (1.96)20.053(1 - 0.053)/(0.03)^2$$

214.3~214

The following assumptions were made during sample size calculation

Z = Standard deviation of the normal distribution = 1.96 (confidence level at 95%)

P = prevalence 5.3% (a cross-sectional study was conducted among pregnant women in Gondar in 2008. Of 209 mothers included in the study, 11 (5.3%) were found to be HBsAg positive.

100-P = pregnant women who not exposed

d = Tolerable error / level of significance = 3.0%.

X = 10% non-respondent rate = 21.43

Sample size = n (Minimum sample size) + X (non- respondent) Sample size (N) = 214 + 22 = 236. Sample size was 236.

## 2.5. Sampling procedure

Study subjects were selected by systematic sampling method by dividing the sample size by the number of pregnant women attending ANC two months before the study period. They were permitted to enter into the ANC clinic room for their routine follow up based on their turn of registration one by one. The aim of the study was briefed to the subjects, and they were asked for their willingness to be interviewed. Those who met the inclusion criteria and volunteer were included. Pregnant women who attended the ANC clinic for more than one time during the study period were excluded.

## 2.6. Study variables

**2.6.1. Dependent variables.** The prevalence of Hepatitis B surface Antigen, among pregnant women attending ANC in DURH, southern Ethiopia.

**2.6.2. Independent variables.** Socio demographic variables like;

1. Maternal age

2. Marital status,

3. Occupational status

4. Educational status

5. Any surgical procedure

6. Gestational age

7. Body tattooing

8. Genital mutilation

9. History of blood transfusion

10. History of multiple sexual practices

## 2.7. Ethical considerations

IRB of Dilla University had approved the ethical clearance. Based on the objective of the study an official letter was sent to Hawassa University Referral Hospital that was involved in the study from Dilla University, College of Health Science Research and Publication committee prior to the data collection period. Confidentiality was maintained and all respondents' questionnaire anonymously prepared. Moreover, informed consent was employed to respondents by explaining the purpose of the study as well as maintaining subject's confidentiality.

## 2.8. Data collection

The data for the study was derived from serological testing and questionnaires. Socio-demographic and associated factors data were collected using a standard structure questionnaire by

professional Nurses after having received a clear explanation of the objective of the study and having signed the informed consent from the participants using a standard consent form designed for this study and interviewed using a questionnaire. Actual data (blood sample and questionnaire) were collected from December 01 to May 28; 2017.

## 2.9. Specimen collection and processing

After obtaining informed consent, 5 ml of venous blood was collected in plane tubes under aseptic conditions from peripheral vein by experienced laboratory personnel from all consenting pregnant women consecutively. These tubes was labeled with unique identification number and processed at the time of collection. The blood samples taken from the individuals were centrifuged at 3000 revolution per minute (RPM) for at least 20 minutes at room temperature and the serum was separated and tested.

## 2.10. Laboratory testing

All the serum samples were tested for HBsAg by using rapid test kit following standard operation procedure.

## 2.11. Quality assurance

The questionnaire was first prepared in English and translated back to Amharic then translated back to English to ensure consistency of the questions. Pre-testing of 5% the questionnaire, which was 12, was done prior to the study. The clarity, understandability and flow of each question and the time to fill the questionnaire was assessed. Daily all the collected data was checked for completeness by the principal investigator. All the data were double entered to ensure the data quality.

## 2.12. Quality control of serological test

Blood samples were collected aseptically from pregnant women and properly labeled by the patient identification number. The specimen was collected by the trained Laboratory personnel. The samples were centrifuged; the serum was separated appropriately and stored until transported to the laboratory.

## 2.13. Data analysis

SPSS version 20 software was used for data analysis. Logistic regression and 95% confidence intervals were calculated to assess the presence and degree of association between independent and outcome variables.

# 3. Results

## 3.1. Socio demographic characteristics

In this study a total of 215 pregnant women have participated with response rate of 91.1%. The mean age of the study participants was 24.76. The majority of study participants were primary school completed, 100 (26.5%), followed by secondary school completed 54 (25%). With regard to parity, majority were multiparous 192 (89.8%). In this study, most of the participants were attending ANC for their second gravidity 90(41.9%) and the remaining were great multipara 23 (10.2%). "Table 1"

**Table 1. Socio-demographic characteristics of pregnant women attending antenatal care at DURH from December to May 28, 2017.**

| Variable | Option | Total no% |
|---|---|---|
| Age in years | 16–20 | 38(17.6%) |
| | 21–25 | 93(43.1%) |
| | 26–30 | 73(33.8%) |
| | 31-above | 11(5.1%) |
| Educational status | illiterate | 45(20.8%) |
| | Primary | 100(46.3%) |
| | Secondary | 54(25.8%) |
| | College | 16(7.4%) |
| Occupation | unemployed | 9(4.2%) |
| | House wife | 142(65.07% |
| | governmental | 28(13%) |
| | Private | 35(16.7%) |
| Ethnicity | Gedio | 100(46.5%) |
| | Wolyta | 38(17.6%) |
| | gurage | 25(11.6%) |
| | amhara | 18(8.3%) |
| | others | 34(15.7%) |
| Religion | protestant | 148(68.5%) |
| | orthodox | 45(20.8%) |
| | Muslim | 20(9.3%) |
| | Others | 2(0.9%) |

## 3.2. Prevalence of HBV infection

From a study conducted among pregnant women attending ANC in DURH, the prevalence of HBsAg by the rapid test was found to be 11 (5.1%). Regarding the age of the participants in the study, the most affected age group were age 26–30 7 out of 11 cases which were 63.3%. Gedeo ethnic group was the predominant (63.6%), and most of the participants were house-wife (81.8%). "S1 Table".

## 3.3. Factors associated with the HBV infection

Factors associated with the prevalence of HBsAg were also determined by taking the proportion of HBsAg detection for the participants in the study.

Of the 215 pregnant women tested for HIV antibody, 10(4.5%) were positive out of which 4 (1.86%) of the study participants who were HIV positive were also positive to HBsAg. There was significant association between HIV infection status and HBV prevalence, at the P-value of 0.019.

Of the study participants, 26 (12.06%) had a history of multiple partners of which 7(3.26%) were positive for HBsAg. The majority of the participants in the study had genital mutilation, 61(28.3%), of which 6(2.7%) were positive for HBsAg. Tattooing was the second prevalence risk factor in study, 34(15.8%), of which 6(2.79%) were positive for HBsAg.

Among those factors affecting the transmission of HBV infection like age, marital status, gravidity, educational level, religion, and ethnicity none has a significant association. But factors like multiple partnership, genital mutilation, abortion history, HIV coinfection had significant association with binary logistic regression P value less than 0.2.

**Table 2. Seroprevalence and associated risk factors among pregnant women in DURH December-May 2017.**

| variable | Option | HBV Status | | | COR(95%CI) | P value |
|---|---|---|---|---|---|---|
| | | Total | Negative | Positive | | |
| Multipartener | Yes | 26(12.06%) | 19(8.8%) | 7(3.2%) | 17.03(4.570–63.537) | 0.00 |
| | No | 189(87.9%) | 185(91.2%) | 4(1.86%) | | |
| Abortion | Yes | 11(5.11%) | 8(3.7%) | 3(1.39%) | 3.56(1.023–12.147) | 0.046 |
| | No | 204(94.8%) | 196(91.1%) | 8(3.7%) | | |
| tattooing | Yes | 34(15.8%) | 28(13.02%) | 6(2.79%) | 7.543(2.157–26.380) | 0.002 |
| | No | 181(84.1%) | 176(81.8%) | 5(2.3%) | | |
| Genital mutilation | Yes | 62(28.84%) | 56(26.04%) | 6(2.79%) | 3.12(0.093–1.07) | 0.065 |
| | No | 153(71.1%) | 148(68.843.7%) | 5(2.3%) | | |
| HIV status | Yes | 10(4.6%) | 4(1.86%) | 6(2.7%) | 18.762(4.302–81.822) | 0.00 |
| | No | 205(95.4%) | 198(92.1%) | 7(3.26%) | | |

In this study, those who had Multiple partner were 17.03 times higher risk to be infected with HBV infection in relation with who had no history of multiple partner with the AOR 17.03 95% CI(4.570–63.537) at P-value 0.00 and those who had history of genital mutilation were 3.17 times at higher risk than who did not had history of genital mutilation with AOR 3.12 95% CI(0.093–1.07) at P-value 0.065. By the same token, tattooing had 7.543 times risks of acquiring the HBV with the AOR of 7.543(2.157–26.380) at the P-value of 0.002, abortion 3.56 times AOR 3.56 95 CI (1.023–12.147) at the P-value of 0.046 and HIV coinfection 18.762 times higher risks of being positive for HBsAg with the AOR 18.762 95% CI (4.302–81.822) at the P-value of 0.00 than their counterparts. "Table 2"

### 3.4. Multi variate analysis for selected factors associated with HBV infection

From this research conducted in DURH showed that five variables are associated with HBV infection by bivariate analysis from this factors only multiple partner and HIV coinfection had been independently associated with HBV infection. Multi partnership showed to be 9.910 times highly associated with HBV infection AOR 9.910 95% CI(1.852–53.103) P-value 0.007. HIV coinfection had been also independently associated with HBV infection, the data showed it was 18.762 times higher in the risk of developing HBV infection AOR 18.762 95% (1.253–55.928) at P-value of 0.030. "Table 3"

## 4. Discussion

In this study, the prevalence of Seropositivity for Hep B virus was found to be 5.1%, which was higher than research conducted in the United States to determine the seroprevalence of hepatitis B surface antigen in pregnant women. In the study conducted in the USA the HBsAg prevalence among white non-Hispanics was 0.60%, black non-Hispanics 0.97%, Hispanics 0.14%, and Asians 5.79% [24], but also much higher than another study conducted in India in 2016, which was 1.01% [25].

**Table 3. Multi variate analysis for selected factors associated with HBV infection, DURH December-May 2017.**

| Variable | Positive | Negative | AOR(95%CI) | P-VALUE |
|---|---|---|---|---|
| Multi partners | 7(3.2%) | 19(8.8%) | 9.910(1.852–53.103) | 0.007 |
| HIV Status | 6(2.7%) | 4(1.86%) | 18.762(1.253–55.928) | 0.030 |

The Seropositivity rate in this study is lower than the recent studies done in Gambia, Ghana, and Southern and Northern parties of Ethiopia, which were 9.2%, 7.7%, 7.3%, and 9.2% respectively (17–20). This is evidenced by the recent publication on the Journal of American Medical Association (JAMA), which highlights the persistent increments of HBV Seropositivity despite the availability of policy and guidelines in screening and vaccinating as per the guideline [10]. This will be a huge assignment for all the stakeholders working toward the control of this virus.

In this study, most of the participants were in the second gravidity 90(41.9%) which was comparable to research conducted in the United States [24, 26].

This disparity might be because of differences in sampling procedures, geographical location government attention level for the infection, Health seeking behaviors of the pregnant mother, Cultural practices, and socioeconomic level differences that need to be justified by further studies.

In this study prevalence of HBV infection, 5.1%, was almost comparable to a similar study conducted in Gondar (5.3%). The most commonly affected age group was age 26–30 in our study which is different from the study conducted in Gondar 16–28 [23].

In this study, the prevalence of HBV infection, 5.1%, was higher than a study conducted in Arbaminch hospital (4.3%). Commonly affected age group were comparable in the two studies in our 26–30 and Arbaminch hospital up of 25–29 years. In this study multi-partners, and HIV status had significant association but in Arbaminch hospital, none of the Sociodemographic factors were significantly associated with HBsAg Seropositivity [27, 28].

The leading risk factors for the acquisition of the virus in this study were genital mutilation 61(28.3%) followed by tattooing 34(15.8%) but it was sharp material in Debra-Tabor hospital, Gondar (93.3%) [23]. This could be the cultural differences between the two areas.

In this study, HIV co-infection and multiple partners had been independently associated with HBV infection, AOR 9.910 95% CI (1.852–53.103) p-value 0.007 and HIV co-infection AOR 18.762 95% CI (1.253–55.928) respectively. This is almost comparable with the studies done in other areas and heralds the due attention needed to tackle this virus in the aforementioned groups of patients [23, 27, 28]. This needs to be verified by further studies. As most of the deliveries are home delivery, the likelihood of being exposed to HBV increases. HIV and HBV share the same mechanisms for transmission and their co-occurrence might tell us these. The negative health outcomes of these dual infections further fuel chronicity and complications [29–33]. These findings need to be verified by further studies for evidence-based interventions.

## 5. Conclusion

This study revealed the prevalence of Seropositivity for Hep B virus by the rapid test was an intermediate prevalence according to the WHO classification, which was found to be 5.1%. So routine screening of pregnant mother for HBsAg during ANC follow up and prompt vaccination of the exposed newborn is recommended.

## Supporting information

**S1 Questionnaire.**
(PDF)

**S1 Table. Seroprevalence of HBsAg among pregnant women attending ANC in DURH from December to May 28, 2017.**
(PDF)

## Acknowledgments

The authors acknowledge all the respondents who took part in the study.

## Author Contributions

**Conceptualization:** Adugnaw Atnafu Atalay, Reta Kassa Abebe, Aberash Eifa Dadhi, Worku Ketema Bededa.

**Data curation:** Adugnaw Atnafu Atalay, Reta Kassa Abebe, Aberash Eifa Dadhi, Worku Ketema Bededa.

**Formal analysis:** Adugnaw Atnafu Atalay, Reta Kassa Abebe, Aberash Eifa Dadhi, Worku Ketema Bededa.

**Funding acquisition:** Adugnaw Atnafu Atalay, Reta Kassa Abebe, Aberash Eifa Dadhi, Worku Ketema Bededa.

**Investigation:** Adugnaw Atnafu Atalay, Reta Kassa Abebe, Aberash Eifa Dadhi, Worku Ketema Bededa.

**Methodology:** Adugnaw Atnafu Atalay, Reta Kassa Abebe, Aberash Eifa Dadhi, Worku Ketema Bededa.

**Project administration:** Adugnaw Atnafu Atalay, Reta Kassa Abebe, Aberash Eifa Dadhi, Worku Ketema Bededa.

**Resources:** Adugnaw Atnafu Atalay, Reta Kassa Abebe, Aberash Eifa Dadhi, Worku Ketema Bededa.

**Software:** Adugnaw Atnafu Atalay, Reta Kassa Abebe, Aberash Eifa Dadhi, Worku Ketema Bededa.

**Supervision:** Reta Kassa Abebe, Aberash Eifa Dadhi, Worku Ketema Bededa.

**Validation:** Adugnaw Atnafu Atalay, Reta Kassa Abebe, Aberash Eifa Dadhi, Worku Ketema Bededa.

**Visualization:** Adugnaw Atnafu Atalay, Reta Kassa Abebe, Aberash Eifa Dadhi, Worku Ketema Bededa.

**Writing – original draft:** Adugnaw Atnafu Atalay, Reta Kassa Abebe, Aberash Eifa Dadhi, Worku Ketema Bededa.

**Writing – review & editing:** Adugnaw Atnafu Atalay, Reta Kassa Abebe, Aberash Eifa Dadhi, Worku Ketema Bededa.

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
