## [Decision Letter · Decision Letter 0]

13 Jan 2021

PONE-D-20-36836

SEROPOSITIVITY OF HEPATITIS B VIRUS AMONG PREGNANT WOMEN ATTENDING ANTENATAL CARE IN DILL UNIVERSITY REFERRAL HOSPITAL ,2017; Health facility based Cross-sectional Study

PLOS ONE

Dear Dr. Bededa,

Thank you for submitting your manuscript to PLOS ONE. After careful consideration, we feel that it has merit but does not fully meet PLOS ONE’s publication criteria as it currently stands. Therefore, we invite you to submit a revised version of the manuscript that addresses the points raised during the review process.

We look forward to receiving your revised manuscript.

Kind regards,

Frank T. Spradley

Academic Editor

PLOS ONE

2. Please include additional information regarding the survey or questionnaire used in the study and ensure that you have provided sufficient details that others could replicate the analyses. For instance, if you developed a questionnaire as part of this study and it is not under a copyright more restrictive than CC-BY, please include a copy, in both the original language and English, as Supporting Information.  If the original language is written in non-Latin characters, for example Amharic, Chinese, or Korean, please use a file format that ensures these characters are visible.

3. Please state whether you validated the questionnaire prior to testing on study participants. Please provide details regarding the validation group within the methods section.

4. In your Methods section, please provide additional information about the participant recruitment method and the demographic details of your participants. Please ensure you have provided sufficient details to replicate the analyses such as:   

-    a description of any inclusion/exclusion criteria that were applied to participant recruitment,

-    a table of relevant demographic details,

-    a statement as to whether your sample can be considered representative of a larger population,

-    a description of how participants were recruited, and

-       descriptions of where participants were recruited and where the research took place.

5. For more information on PLOS ONE's expectations for statistical reporting, please see https://journals.plos.org/plosone/s/submission-guidelines.#loc-statistical-reporting. Please update your Methods and Results sections accordingly.

6. Please include your tables as part of your main manuscript and remove the individual files. Please note that supplementary tables (should remain/ be uploaded) as separate "supporting information" files

7. Please amend either the title on the online submission form (via Edit Submission) or the title in the manuscript so that they are identical.

Reviewers' comments:

Reviewer's Responses to Questions

**Comments to the Author**

1. Is the manuscript technically sound, and do the data support the conclusions?

Reviewer #1: Yes

Reviewer #2: Yes

2. Has the statistical analysis been performed appropriately and rigorously? 

Reviewer #1: Yes

Reviewer #2: I Don't Know

3. Have the authors made all data underlying the findings in their manuscript fully available?

Reviewer #1: Yes

Reviewer #2: No

4. Is the manuscript presented in an intelligible fashion and written in standard English?

Reviewer #1: No

Reviewer #2: No

5. Review Comments to the Author

Reviewer #1: This is an interesting study aimed at assessing the seroprevalence and associated risk factor of HBV among pregnant women attending Antenatal Care (ANC) in Dilla University Referral Hospital (DURH), Southern Ethiopia. There are a number of issues in the paper.

Abstract:

Methods: The authors should include the inclusion and exclusion criteria in the abstract. What were the outcome measures?

The authors wrote: 215 study participates were tested for HIV antibody, of which 10 (4.5%) were turned

to be positive for HIV. The authors should avoid starting a sentence with figures.

The authors stated: The Seropositivity of Hepatitis B Virus among Pregnant Women was found to be significant. Why did they say that the study is significant? How did they arrive at significant?

iNTRODUCTION

It is well written.

The authors should beef up the justification for the study.

Results and Discussion are ok.

Reviewer #2: The manuscript entitled: "SEROPOSITIVITY OF HEPATITIS B VIRUS AMONG PREGNANT WOMEN ATTENDING ANTENATAL CARE IN DILL UNIVERSITY REFERRAL HOSPITAL ,2017" is a facility based cross-sectional study that aimed to assess the seroprevalence of HBV among pregnant women attending antenatal care in DURH as well as the risk factors associated with HBV infection in this population. Overall, this study presents the results of original scientific research that addresses an interesting question and was designed appropriately to achieve the two aforementioned aims. However, the manuscript presents certain issues and requires substantial revision in order to be published. Firstly, in the Introduction, the authors should clearly summarize the state of research in the field by citing more recent seroplevalence studies, globally. Since the author's goal is to underline the need for global screening and vaccination during pregnancy, they should consider reading and citing the latest guidelines from WHO, EASL, or AASLD, like the article: "US Preventive Services Task Force; Douglas K Owens et al. Screening for Hepatitis B Virus Infection in Pregnant Women: US Preventive Services Task Force Reaffirmation Recommendation Statement, JAMA. 2019 Jul 23;322(4):349-354." In the Materials and Methods, the sample size calculation procedure is not easy to read and understand, therefore it should be elucidated. One major issue is the Results section of the manuscript, as the results are not clearly presented, some values are missing and some values are different for the ones in the Tables. Overall, the authors should pay the attention to rephrase the results in an appropriate manner. A Table summarizing the sociodemographic characteristics should also be added, while the other Tables could also be restructured. Overall, the discussion is interesting, as it compares the results with previous data from the same country and two studies conducted in the USA and in India. However, the authors should cite the original studies from these two countries and enrich the discussion with more recent publications. Finally, since the authors' mother tongue is not English, they should carefully revise the manuscript for grammatical and syntactical errors.

6. PLOS authors have the option to publish the peer review history of their article (what does this mean?). If published, this will include your full peer review and any attached files.

Reviewer #1: **Yes: **George Eleje

Reviewer #2: **Yes: **Nikoletta Maria Tagkou

---

## [Author Response · Author response to Decision Letter 0]

12 Feb 2021

Manuscript PONE-D-20-36836

Point- by- Point Rebuttal Letter

We really thank the editor and the two reviewers for their valuable comments on our manuscript.

Please kindly find below our response to each point raised by the academic editor and reviewers. We hope that we clearly addressed all of them, and that the manuscript will be now suited for publication. We bolded the comments, and highlighted the responses by water blue color.

Sincerely,

On behalf of all the four authors,

Worku Ketema Bededa

Academic editor:

Journal requirements

1. Plos one templates

We have checked the templates and made the adjustments to meet the journal requirements.

2. Additional information regarding survey or questionnaire-

We put the questionnaire in the supplementary data in both Amharic and English and please kindly see the supplementary documents.

3. State whether you validated the questionnaire prior to testing on the study participants. Please provide details regarding the validation groups within the method section

Thank you for your comments; The questionnaire was first prepared in English and translated back to Amharic then translated back to English to ensure consistency of the questions. Pre-testing was done on 12 pregnant mother prior to the study (5 % of the total sample size, which was 236). The clarity, understandability and flow of each question and the time to fill the questionnaire were assessed with a bit modification of the questionnaire. All the collected data was checked for completeness by the principal investigator daily. All the data were double entered to ensure the data quality

4. In your methods; Please provide additional information about the participant recruitment method and the demographic details of your participants. Please ensure you have provided sufficient details to replicate the analysis such as;

-A description of any inclusion/exclusion criteria that were applied to participant recruitment

-A table of relevant demographic details

-A statement as to whether your sample can be considered as representative of a large population

- Description of how participants were recruited, and

-Description of where participants were recruited and where the research took place

Response; We are grateful to respond to your constructive comments, and here are the responses one by one

Regarding the Eligibility criteria

Inclusion Criteria

All Pregnant women were attending antenatal clinic in the DURH during the study period and who were volunteering and give informed consent

Exclusion Criteria

Those women who were unable to communicate due to any problem

The table of relevant demographic details has been incorporated in the main manuscript(Table 1)

One of the limitations of this paper is small sample size( due to financial constraint by the time of the study), and I admit that the finding will not be an exact representative of large population, but will only give as an insight on the burden of the virus and associated factors. It will pave a path for the future study that will be done with better sample size.

Participant recruitment has been elaborated under the sampling procedure. Participants were recruited from the ANC clinic of Dilla University Referral Hospital, Dilla, Ethiopia

Sampling procedure

We appreciate for your review, and here is the response for this specific comment; Study subjects were selected by systematic sampling method by dividing the sample size by the number of pregnant women attending ANC two months before the study period. They were permitted to enter into the ANC clinic room for their routine follow up based on their turn of registration one by one. The aim of the study was briefed to the subjects, and they were asked for their willingness to be interviewed. Those who met the inclusion criteria and volunteer were included. Pregnant women who attended the ANC clinic for more than one time during the study period were excluded.

Study variables-We are sorry for not to incorporate this in the original manuscript initially,and here are the edited one;

Dependent variables

The prevalence of Hepatitis B surface Antigen, among pregnant women attending ANC in DURH, southern Ethiopia

Independent variables

Socio demographic variables like;

1. Maternal age marital status,

2. Occupational status

3. Educational status

4. Any surgical procedure

5. Gestational age

6. Body tattooing

7. Genital mutilation

8. History of blood transfusion

9. History of multiple sexual practices

6. Table as part of main manuscript and Supplementary table- Thank you for the comments and we have made corrections.

7. Title differences- Thank you, we edited it

Reviewer #1

Comment 1

Abstract

Methods; the authors should include the inclusion and inclusion criteria in the abstract. What were the outcome measures?

Response 1; the reviewer has made interesting points, and we made an amendments.

All Pregnant women who were attending antenatal clinic and were volunteer during the study period were included, whereas those women who were unable to communicate due to any problem and not volunteer to give informed consent were excluded.

Comment 2; the authors wrote; 215 study participates were tested for HIV antibody, of which 10 (4.5%) were turned to be positive for HIV. 4 (1.86 %) of the study participants who were tested for HIV were also positive for HBsAg. The authors should avoid starting a sentence with figures.

Response 2; we thank the reviewer for his assessment, and we edited is as below

Among the study participants, 91.1 % (215) were tested for HIV antibody during the ANC visit, with the positivity rate of 4.5 %. The result showed 1.86 % of the study participants who were tested for HIV were also positive for HBsAg.

Comment 3; the authors stated the Seropositivity of Hepatitis B Virus among Pregnant Women was found to be significant. Why did they say that the study is significant? How did they arrive at significant?

Response 3; we said it was significant in our conclusion, after comparing our result with the WHO established criteria 2015(2).

Comment 4; the authors should beef up the justification for the study

Response 4; we thank the reviewer for constructive comments,

Some amendments have been done, at the last paragraph of the introduction. We especially interested to quote the statement word by word, “Although there are guidelines for universal infant HBV vaccination, rates of maternal HBV infection have increased annually by 5.5% since 1998” (10).We ,hence interested to investigate the burden of Seropositivity in the specific study area, and put the path for the possible interventions if needed.

Reviewer # 2

Comment 1

In the introduction part; the authors should clearly summarize the state of research in the field by citing more recent seroprevalence studies, globally. From JAMA, WHO…

Response 1; we understand and agree with this observation, and we thank the reviewer for pointing this out. We incorporated the latest publications on the seroprevalence, among which the one published on JAMA. We also added the recent studies done globally and locally like studies done in Gambia, Ghana, Southern Ethiopia, and Northern Ethiopia, included in the introduction and discussion parts.

Comment 2; in the material and method, the sample size calculation procedure is not easy to read and understand, therefore it should be elucidated

Response 2; we apologise for the inconveniences and we hope know you will get the documents more visible

Sample was calculated by taking overall Hepatitis B infection prevalence among a cross-sectional study conducted among pregnant women in Gondar in 2008. Of 209 mothers included in the study, 5.3% 3.0% level of significance / margin of error (23). .This sample size will be estimated using the formula for calculating sample size for cross sectional study of estimation a single population proportion as described below.

n = (Zα/2)2 P (1-P)/ (d) 2

= (1.96)20.053(1-0.053)/ (0.03)2

214.3~214

Assumptions

Z =Standard deviation of the normal distribution = 1.96 (confidence level at 95%)

P = prevalence 5.3 %( a cross-sectional study was conducted among pregnant women in Gondar in 2008.

100-P = pregnant women who not exposed

d = Tolerable error / level of significance = 3.0%.

X=10 % non-respondent rate = 21.43

Sample size = n (Minimum sample size) + X (non- respondent) Sample size (N) = 214 + 22 = 236. Sample size was 236

Comment 3; Result-The results are not clearly presented, some values are missing and some values are different for the one in the tables. Overall the author should pay the attention to rephrase the results in appropriate manner. A table summarizing the sociodemographic characteristics should also be added, while the other tables could also be restructured.

Response 3; we thank the reviewer for pointing out this inconsistency. It is now corrected 

Comment 4; Discussion-The authors should site the original studies from USA and India (mentioned in the discussion), and enrich the studies with more recent publications.

Response 4; we thank the reviewer for his kind comments and useful insights.

We tried to incorporate the recent publications on the specific topic, and also reviewed and incorporated the original studies done in USA and India that have mentioned in the original manuscript. In this study, the prevalence of Seropositivity for Hep B virus was found to be 5.1%, which was higher than research conducted in the United States to determine the seroprevalence of hepatitis B surface antigen in pregnant women. HBsAg prevalence among white non-Hispanics was 0.60%, black non-Hispanics 0.97%, Hispanics 0.14%, and Asians 5.79% (20) .It also much higher than another study conducted in India in 2016, which was 1.01 % (21) .In this study, most of the participants were in the second gravidity 90(41.9%) which was comparable to research conducted in the United States .

The Seropositivity rate in this study is also lower than the recent studies done in Gambia, Ghana, Southern and Northern parties of Ethiopia, which were 9.2 %, 7.7%, 7.3 % and 9.2 % respectively. (17-20)

Comment 5; since the authors mother tongue is not English, they should carefully revise the manuscript for grammatical and syntactical errors

Response 5; We admit that, and tried to correct some grammatical errors with the help of my fellow English department friends, and I hope you will get it better than the previous documents.

---

## [Decision Letter · Decision Letter 1]

8 Mar 2021

PONE-D-20-36836R1

Seroprevalence of hepatitis B virus among pregnant women attending Antenatal care in Dilla University Referral Hospital Gedio Zone, Ethiopia ; Health facility based Cross-sectional Study

PLOS ONE

Dear Dr. Bededa,

Thank you for submitting your manuscript to PLOS ONE. After careful consideration, we feel that it has merit but does not fully meet PLOS ONE’s publication criteria as it currently stands. Therefore, we invite you to submit a revised version of the manuscript that addresses the points raised during the review process.

We look forward to receiving your revised manuscript.

Kind regards,

Frank T. Spradley

Academic Editor

PLOS ONE

Journal Requirements:

Reviewers' comments:

Reviewer's Responses to Questions

**Comments to the Author**

1. If the authors have adequately addressed your comments raised in a previous round of review and you feel that this manuscript is now acceptable for publication, you may indicate that here to bypass the “Comments to the Author” section, enter your conflict of interest statement in the “Confidential to Editor” section, and submit your "Accept" recommendation.

Reviewer #1: All comments have been addressed

Reviewer #2: (No Response)

2. Is the manuscript technically sound, and do the data support the conclusions?

Reviewer #1: Yes

Reviewer #2: Yes

3. Has the statistical analysis been performed appropriately and rigorously? 

Reviewer #1: Yes

Reviewer #2: Yes

4. Have the authors made all data underlying the findings in their manuscript fully available?

Reviewer #1: Yes

Reviewer #2: Yes

5. Is the manuscript presented in an intelligible fashion and written in standard English?

Reviewer #1: Yes

Reviewer #2: No

6. Review Comments to the Author

Reviewer #1: The authors have endeavoured to revise the manuscript as recommended. They have addressed the most salient issues raised

Reviewer #2: We thank the authors for adequately studying and addressing both the Editor’s and our comments. The manuscript is now closer to meeting PLOS publication criteria, but there are still some points that can be mended.

Abstract and Introduction: there are some minor grammatical mistakes and some parentheses that are out of place, that could easily be corrected.

Material and Method: Thank you for pointing out the inclusion and exclusion criteria and for elucidating the sample size calculation. In the section 2.6.2 Independent variables, maternal age and marital status should be in different lines, as they represent two separate sociodemographic characteristics.

Results: The authors should especially pay attention at further correcting section 3.3. For example, they state “955 CI” or “95 CI” where there should be 95% CI. In the text they are referring to sharp material sharing with others, while there are no relevant data in Table 2. Concerning genital mutilation, the p-value is different in the text and in Table 2. Also, the last line from Table 2, HIV status- Option No, is missing.

Discussion: There are some parentheses here also that are misplaced.

The authors specifically state “In this study multi-partners, gravidity and

HIV status had significant association”, while earlier, in the results, they state that “Among those factors affecting the transmission of HBV infection like age, marital status, gravidity, educational level, religion, and ethnicity none has a significant association”. Since gravidity was not found to have a significant association with HBsAg status, this sentence should be corrected.

Finally, when the authors say that HIV and HBV share the same mechanisms of transmission, the should cite relevant bibliography.

When this issues are also addressed, then the manuscript will be finally acceptable for publication.

7. PLOS authors have the option to publish the peer review history of their article (what does this mean?). If published, this will include your full peer review and any attached files.

Reviewer #1: **Yes: **George Eleje

Reviewer #2: **Yes: **Tagkou Nikoletta Maria

---

## [Author Response · Author response to Decision Letter 1]

11 Mar 2021

I,on behalf of all authors,am really thankful to respond to your valuable and constructive comments,and really appreciate your reasonable and scientific comments! Thank you!

---

## [Editor Report · Decision Letter 2]

15 Mar 2021

Seroprevalence of hepatitis B virus among pregnant women attending Antenatal care in Dilla University Referral Hospital Gedio Zone, Ethiopia ; Health facility based Cross-sectional Study

PONE-D-20-36836R2

Dear Dr. Bededa,

We’re pleased to inform you that your manuscript has been judged scientifically suitable for publication and will be formally accepted for publication once it meets all outstanding technical requirements.

Kind regards,

Frank T. Spradley

Academic Editor

PLOS ONE

---

## [Editor Report · Acceptance letter]

17 Mar 2021

PONE-D-20-36836R2 

Seroprevalence of hepatitis B virus among pregnant women attending Antenatal care in Dilla University Referral Hospital Gedio Zone, Ethiopia; Health facility based Cross-sectional Study 

Dear Dr. Bededa:

I'm pleased to inform you that your manuscript has been deemed suitable for publication in PLOS ONE. Congratulations! Your manuscript is now with our production department. 

Kind regards, 

on behalf of

Dr. Frank T. Spradley 

Academic Editor

PLOS ONE